# *Gelsemium elegans* Benth: Chemical Components, Pharmacological Effects, and Toxicity Mechanisms

**DOI:** 10.3390/molecules26237145

**Published:** 2021-11-25

**Authors:** Hailing Lin, Hongqiang Qiu, Yu Cheng, Maobai Liu, Maohua Chen, Youxiong Que, Wancai Que

**Affiliations:** 1Department of Pharmacy, Fujian Medical University Union Hospital, 29 Xin Quan Rd, Gulou, Fuzhou 350001, China; Iricehl@163.com (H.L.); honjohn@126.com (H.Q.); 86cy@163.com (Y.C.); liumb0591@sina.com (M.L.); 13075825281@163.com (M.C.); 2College of Agriculture, Fujian Agriculture and Forestry University, Fuzhou 350002, China

**Keywords:** *Gelsemium elegans* Benth (GEB), pharmacology, toxicology, mechanisms

## Abstract

*Gelsemium elegans* Benth (GEB), also known as heartbreak grass, is a highly poisonous plant belonging to the family *Loganiaceae* and genus *Gelsemium* that has broad application prospects in medicine. This article reviews its chemical components, pharmacological effects, toxicity mechanisms, and research progress in clinical applications in recent years. Indole alkaloids are the main active components of GEB and have a variety of pharmacological and biological functions. They have anti-tumor, anti-inflammatory, analgesic, and immunomodulation properties, with the therapeutic dose being close to the toxic dose. Application of small-dose indole alkaloids fails to work effectively, while high-dose usage is prone to poisoning, aggravating the patient’s conditions. Special caution is needed, especially to observe the changes in the disease condition of the patients in clinical practice. In-depth research on the chemical components and mechanisms of GEB is essential to the development of promising lead compounds and lays the foundation for extensive clinical application and safe usage of GEB in the future.

## 1. Introduction

*Gelsemium elegans* Benth (GEB), also known as heartbreak grass, is a highly poisonous plant belonging to the family *Loganiaceae* and genus *Gelsemium*. GEB has been made into a galenic formulation in other countries and is included in the American Drug Index [1]. In China, herbal classics, the Shennong’s Materia Medica Classic (Shennong Bencao Jing) [2], and the Compendium of Materia Medica (Bencao Gangmu) [3] also record that due to its relatively high toxicity, GEB is used externally to dispel the wind, reduce swelling, remove poison, kill insects, and relieve itching. In the 1930s, Chinese scholars began to study GEB from the perspectives of biochemistry [4,5,6,7], pharmacology [8,9,10], toxicology [11], and genomics [12,13] Given its relatively high toxicity, the clinical application of GEB has been limited. In the 1980s, the scope of clinical application of GEB was expanded and applied for anti-tumor [14], anti-inflammatory and analgesic [15], immunomodulation [16], and various biological functions.

In the present study, the searches and reviews were conducted from the databases of MEDLINE/PubMed, Google Scholar, CNKI, Scopus, and Web of Science on 18 September 2020. With the search terms of *G. elegans* both in English and Chinese, 135 papers were retrieved after the first search, and 60 papers among them which were tightly related with chemical components, pharmacological effects, and toxicity mechanisms were identified for collection and further review. The corresponding research progress of GEB and its recent clinical application were then summarized (see Figure 1 for details). The present review aims to promote the development of basic and applied research on GEB and the development of lead compounds based on the chemical components of GEB.

## 2. Chemical Components of GEB

The main active components of GEB are indole alkaloids comprising six categories: Sarpagine, methyl gelsedine, gelsemine, humantenine, koumine, and yohimbane [7] (Figure 2). The extract of GEB contains nearly 100 alkaloids in these six categories, with the highest contents of koumine, followed by gelsevirine, gelsemine, humantenine, and gelsenicine [7]. In addition, GEB contains a variety of iridoid glycosides, iridoid, steroids, and other non-alkaloid compounds [17]. However, the chemical components of GEB from different origins are different [18].

## 3. Pharmacological Effects of GEB

### 3.1. Clinical Pharmacology of GEB

#### 3.1.1. Anti-Tumor Effects

Different indole alkaloids of GEB obtained by various extraction methods have anti-tumor effects in vitro and in vivo. Among them, koumine has the strongest anti-tumor effect possibly through blocking the cell cycle and promoting apoptosis [19]. Huang et al. [19] inoculated H22 mouse liver cancer cells to prepare a tumor-bearing animal model, and it showed that the anti-tumor effect of koumine was better than gelsemine and 1-methoxygelsemine by acridine orange (fluorescent cationic dye)/ethidium bromide dual staining. Koumine induced the apoptosis of SW480 human colon cancer cells and blocked the S/G2 transition in the cell cycle. In addition, koumine, gelsemine, gelsenicine, and 1-methoxygelsemine demonstrated anti-tumor activities in the digestive tract in a dose-dependent manner, showing a certain structure-activity relationship [20]. The study by Chi et al. [21] revealed that koumine has significant anti-tumor effects in vitro and in vivo, not only inhibiting the proliferation of a variety of common cell lines, but also having significant inhibitory effects on transplanted tumors in mice. It is interesting that the inhibitory effect of koumine on rectal cancer leads to the activation of the apoptotic pathway by reducing expression of the B-cell lymphoma-2 (Bcl-2) gene. In addition, the reduced expression of Bcl-2 is related to the accumulation of rabbit anti-human BAX, anti-cytochrome oxidase, and caspase-3 [22,23]. Moreover, what should be stressed here is that, gelsebanine extracted from the stems and leaves of GEB, reportedly has strong cytotoxic effects on the A549 human lung adenocarcinoma cell line [24]. Furthermore, gelsedine and other alkaloids of GEB have strong cytotoxic effects on the A431 human epidermoid carcinoma cell line [25]. These studies showed that GEB has inhibitory effects on the proliferation of liver cancer, colon cancer, gastric cancer, rectal cancer, lung adenocarcinoma, and epidermoid carcinoma cells (Table 1), and its main component is all alkaloids. These findings lay an experimental foundation and guide the direction for later drug research in this field. However, further analyses of the specific mechanisms, such as the molecular responses from the aspects of metabolome, transcriptome, and proteome, are waiting to be conducted.

#### 3.1.2. Anti-Inflammatory and Analgesic Effects

The total alkaloids and main active components of GEB have certain roles in anti-inflammatory and analgesic treatments, mainly for pain caused by neuropathy, rheumatoid arthritis, skin ulcers, and tumors. However, their analgesic effects are not subject to the development of tolerance [26]. It is worth noting that alkaloids of GEB also have obvious anti-inflammatory effects on analgesic treatment [27]. Tan et al. (1988) evaluated the effects of GEB alkaloids on mice using the hot-plate experiment, acetic-acid-induced writhing test, and tail-flick test (tail–flick test evoked by radiant heat). The results showed that GEB alkaloids increased the pain threshold in mice without causing tolerance compared with morphine [26]. In addition, the combination of GEB alkaloids, pentobarbital sodium, and chloral hydrate were found to enhance the hypnotic effects on the animals [26]. A study by Ling et al. (2014) also revealed that koumine could alleviate neuropathic pain in diabetic rats, without affecting blood sugar and body weight [27]. It significantly reduced the sciatic nerve axon and myelin damage, but accelerated the sensory nerve conduction velocity in the animals [27]. Moreover, research indicated that subcutaneous injection of gelsedine in mice had potential analgesic effects on writhing induced by acetic acid, and nociceptive behavior induced by formalin or thermal hyperalgesia caused by chronic contractile injury [28]. Nevertheless, it had dose-dependent effects on both inflammation and neuropathic pain [28]. Surprisingly, the effective dose 50 (EC_50_) was much lower than the half-lethal dose 50 (LD_50_) (95% confidence interval: 100–200 µg/kg), indicating that gelsedine can be used to reduce inflammation and neuropathic pain and should have good therapeutic safety [28]. It is particularly worrying that, under chronic pain conditions, gelsemine produces strong analgesic effects, without developing tolerance. For the long-term use of analgesia for cancer patients, the research and development of gelsemine should be strengthened and optimized [29]. Total alkaloids of GEB play an anti-inflammatory role in suppressing the synthesis and release of prostaglandins from inflammation [30].

#### 3.1.3. Anti-Anxiety Effects

Low-dose koumine has significant effects on anti-anxiety and does not affect other autonomic nerve and physiological functions. Its mechanisms may not only increase the levels of neurosteroid pregnenolone and allopregnanolone in the hippocampus [31], but can also be related to the agonistic effect on glycine receptors [32]. A low concentration (5–10 µg/kg) of koumine significantly alleviates the cognitive impairment induced by beta amyloid (main neurotoxin of Alzheimer’s disease [AD]), suggesting that koumine inhibits the proliferation of glial cells associated with AD [33]. Koumine inhibits the overexpression of interleukin 6 (IL-6), IL-1, and tumor necrosis factor (TNF) in tumors, showing anti-neuritis effects [34]. These findings lay a certain foundation for the treatment of AD and related neurodegenerative diseases. It is reasonably deduced that koumine inhibits glycogen synthase kinase-3 (GSK3) (activity of GSK3 increases in the brain of AD patients, and GSK3 is a strong promoter of pro-inflammatory cytokines), thereby reducing GSK3-mediated neuroinflammation [33].

#### 3.1.4. Immunomodulatory Effects

Koumine inhibits not only the proliferation of CD4+ T lymphocytes, but also the immune activity of T lymphocytes in vitro [34]. Different concentrations of koumine significantly reduce IL-2 and interferon-c levels and the cytokines of T helper type 1 (Th1) cells, for example, 100 µg/mL koumine significantly increases the levels of IL-10, but not IL-4 in Th2 cells [34]. It is noted that certain concentrations of koumine also induce the differentiation of Th1 cells into Th2 cells through immune–inflammation interaction, exerting anti-psoriatic effects [34]. In addition, koumine reduces the production of IL-1β and TNF-α to improve joint damage, and thus relieves pain caused by arthritis. It also improves white blood cell and erythrocyte sedimentation and inhibits thymus and liver enlargement, which is related to the increase in serum anti-type II collagen antibodies after the immune response [35]. There is a report that koumine inhibits the proliferation of mouse splenocytes induced by mixed lymphocyte culture, 2 µg/mL concanavalin A or 10 µg/mL bacterial lipopolysaccharide (LPS) to varying degrees. In addition, high concentrations of koumine significantly inhibit complement-mediated hemolysis [36]. Koumine not only alleviates the intestinal barrier dysfunction induced by LPS, but also reduces oxidative stress and inflammation induced by LPS through activating the nuclear erythroid 2-related factor 2 (Nrf2)/nuclear factor (NF)-polysaccharide signaling pathway. Moreover, it can alleviate the increase of TNF-α, IL-6, IL-1, nitric oxide, inducible nitric oxide synthase, and cyclooxygenase-2 levels [37].

As a component of medicine, GEB is also used to treat various types of skin diseases including systemic lupus erythematosus, psoriasis, and neurodermatitis [38]. Koumine significantly inhibits the hyperproliferation of epithelial cells in mouse models and promotes the normal keratinization of the epidermis to form a granular layer, indicating its anti-psoriasis effect [38]. In addition, koumine significantly reduces serum IL-2 levels in mice in a dose-dependent manner and participates in cellular immunomodulation [38].

#### 3.1.5. Cardiovascular Repair

A previous study showed that total alkaloids of GEB slow down the heart rate [39] and inhibit the tachycardia caused by adrenaline, which may be related to their inhibitory or blocking effect on the receptors [40,41]. In addition, total alkaloids of GEB weaken myocardial contractility and diastolic blood vessels rapidly [42], which are closely related to their excitatory effect on the cholinergic nerves in the cardiovascular system and the muscarinic receptors in the peripheral nervous system [43].

#### 3.1.6. Hematopoietic Protection

As for hematopoietic protection, total alkaloids of GEB not only promote the production of red blood cells, but also alleviate the leukopenia caused by cyclophosphamide (Cy) chemotherapy to a certain extent [44]. In addition, they reverse the reduction of peripheral blood cells, red blood cells, platelets, bone marrow total nucleated cells caused by Cy chemotherapy and exert protective effects on hematopoietic function [45,46].

#### 3.1.7. Mydriasis

Regarding mydriasis, total alkaloids of GEB have an anticholinesterase-like effect and exert a muscarinic acetylcholine receptor-like effect by exciting the peripheral autonomic nerves [47]. Wang et al. (1990) recruited 69 volunteers and used the GEB alkaloids eye drops to dilate the pupils and regulate the palsy effects on the eyes. At 6 h after treatment, various indicators of the volunteers resumed to normal, and the recovery time of the participants was faster than in those receiving homatropine and tropicamide [48], suggesting that total alkaloids of GEB may be used for the development of a novel mydriatic agent.

### 3.2. Pharmacokinetic Research of GEB

There are few reports on the pharmacokinetics (PKs) of the monomeric compounds of GEB. Xu et al. [49] established a method to determine the contents of koumine in blood and tissues of rats by high-performance liquid chromatography. The results showed that after 5 min of koumine administration, the contents of koumine were widely distributed and highly expressed in the tissue, stomach, intestines, liver, body fat, kidneys, and spleen [49]. In addition, the distribution speed of koumine was fast, and the koumine in most of the tissues reached peak concentrations 5 min after intragastric administration. However, koumine in the small intestine and liver only reached peak concentrations at 15 min after administration, and the koumine contents in most tissues were reduced at 60 min after the administration [49]. Wang et al. [50] used ultra-performance liquid chromatography–tandem mass spectrometry to determine the koumine and gelsemine levels in rat plasma. When the volume of 20 µL was used as the standard, the linear range was 0.1-500 ng/mL for koumine and 0.2–100 ng/mL for gelsemine respectively, while the corresponding lower detection limit was 0.1 ng/mL and 0.2 ng/mL. Another study has also revealed the impact of cytochrome P450 enzymes in liver microsomes of different species of animals on the metabolism of koumine [51]. Liu et al. [52] used two-dimensional liquid chromatography to establish a model for the simultaneous detection of gelsemine, koumine, and gelsenicine. This should provide a theoretical basis for the PK study and toxicological analysis of GEB, as well as the clinical diagnosis and treatment of poisoning.

## 4. Mechanisms of GEB Toxicity

It is generally recognized that the whole plant of GEB is highly toxic, with stronger toxicity in the root and young leaf tissues, and the strongest toxicity in the root bark, which causes poisoning through oral administration [53]. The main toxic components of GEB are indole alkaloids. Among them, koumine content was highest, followed by gelsemine, and gelsenicine has the highest toxicity [54]. GEB poisoning mainly involves damages to the nervous system, the digestive system, and the respiratory systems, with the common symptoms of vomiting, dizziness, abdominal pain, respiratory depression, convulsion, coma, and spasms [55]. The toxicity of GEB is usually presented as the LD_50_ or minimal lethal concentration. Rujjanawate et al. [8] suggested that the LD_50_ of total alkaloids of GEB was 15 mg/kg by oral administration and 4 mg/kg by intraperitoneal injection in mice. In addition, 2–4 mg/kg gelsebanine alleviated anxiety and nerve pain in mice without showing significant toxicity to the animals, indicating that <2 mg/kg gelsebanine may be safe for animals. Table 2 shows the toxicity data of GEB alkaloids and its extracts.

As shown in Table 2, total alkaloids of GEB may cause death in animals or humans, mainly by inhibiting the respiratory system to cause death due to respiratory failure. They may also cause symptoms, such as slowing the heart rate and lowering blood pressure. A study by Yi et al. (2003) showed that injection of >4 mg/kg total alkaloids of GEB caused respiratory depression accompanied by a decreased heart rate and blood pressure in rabbit. When an injection of >8 mg/kg total alkaloids of GEB takes place, it can cause respiratory failure and death in the animals. A previous study revealed that total alkaloids of GEB acted on the vagus nerve and myocardium to cause circulatory disorders and aggravated the damage of the liver and kidneys [60].

## 5. Summary and Prospect

This paper summarizes the research progress of the chemical components, pharmacological effects, and mechanisms of toxicity, as well as the clinical applications of GEB. However, the complex chemical components of GEB and the complicated monomer extraction process together with the low extraction content hinder the research progress of pharmacological effects and the clinical application remained at the level of total alkaloids of GEB. The main component of total alkaloids of GEB is koumine. Nevertheless, the existing reports on koumine have mainly focused on the in vitro studies. Due to the fact that the therapeutic dose of GEB is close to the toxic dose, it is necessary to identify the appropriate dose or the exact chemical components of GEB with high efficiency and low toxicity and to clarify the pharmacological and toxicological effects. In addition, future studies should not only concentrate on the elaboration of the mechanism of GEB causing poisoning in humans and animals, but also aim to understand the specific pharmacology and mechanisms of toxicity of GEB. We do hope that the rapid advancement of research on antitumor, anti-inflammatory, and analgesic aspects of koumine can provide novel therapeutic regimens for the clinical application of this natural medicine.

## Figures and Tables

**Figure 1 molecules-26-07145-f001:**
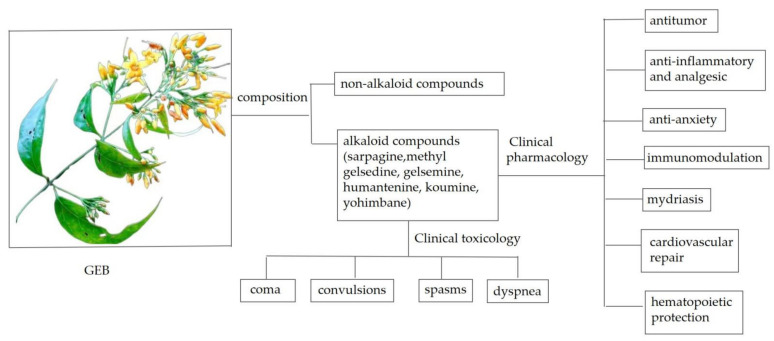
Overview of GEB.

**Figure 2 molecules-26-07145-f002:**
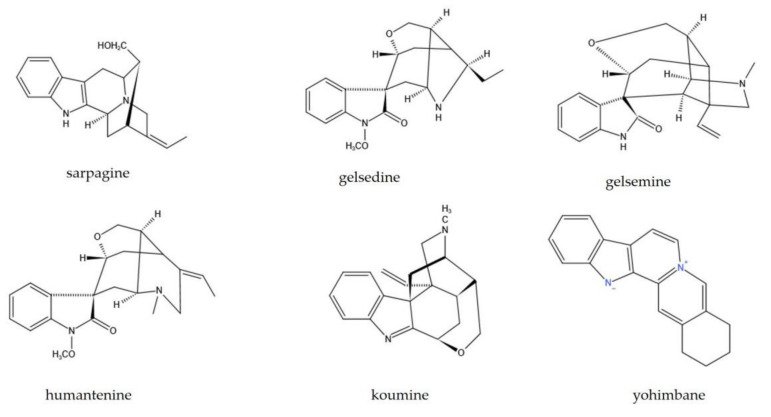
Six structural categories of GEB alkaloids.

**Table 1 molecules-26-07145-t001:** Anti-tumor mechanisms of GEB.

Active Component	Tumor Type	Mechanism	EC_50_(mm)/EC_50_(um)	References
Koumine	H22 liver cancer cells	Inhibits the proliferation of solid tumors derived from H22 in mice		[19]
Koumine	SW480 human colon cancer cells	Induces apoptosis and blocks the S/G2 transitio n in SW480 cells	EC_50_: 0.45 ± 0.10	[20]
Gelsenicine	SW480 human colon cancer cells	Inhibits tumor cell proliferation	EC_50_: 0.52 ± 0.22	[20]
Gelsenicine	MGC80-3 human gastric cancer cells	Inhibits tumor cell proliferation	EC_50_: 1.14 ± 0.23	[20]
N1-methoxygelsemine	SW480 human colon cancer cells	Inhibits tumor cell proliferation	EC_50_: 1.41 ± 0.06	[20]
N1-methoxygelsemine	MGC80-3 human gastric cancer cells	Inhibits tumor cell proliferation	EC_50_: 1.22 ± 0.01	[20]
Koumine	HCT116 human colorectal cancer cells	By reducing the expression of Bcl-2 to activate the apoptotic pathway; the expression of Bcl-2 is related to the increase of BAX, CytC, and caspase-3	EC_50_: 0.07056	[22,23]
Koumine	HT29 human colorectal cancer cells	By reducing the expression of Bcl-2 to activate the apoptotic pathway; the expression of Bcl-2 is related to the increase of BAX, CytC, and caspase-3	EC_50_: 0.06282	[22,23]
Gelsebanine	A549 human lung adenocarcinoma cells	Has strong cytotoxicity in A549 cells	EC_50_: 6.34 × 10^−4^	[24]
6, Gelsenicine	A431 human epidermoid carcinoma cells	Has strong cytotoxicity in A431 cells	EC_50_: 37	[25]
7, Gelsedine	A431 human epidermoid carcinoma cells	Has strong cytotoxicity in A431 cells	EC_50_: 0.35	[25]
8, Gelsemicine	A431 human epidermoid carcinoma cells	Has strong cytotoxicity in A431 cells	EC_50_: 0.75	[25]

**Table 2 molecules-26-07145-t002:** Toxicity data of GEB.

Active Component	Laboratory Animal	Mode of Administration	LD_50_ or MLC Range/(mg/kg)	Poisoning Symptoms	Reference
Total alkaloids of GEB	Mice	Oral administration	LD_50_: 15 (13–17)	Reduced activity, slowed breathing, convulsions, respiratory depression	[56]
Total alkaloids of GEB	Mice	Intraperitoneal injection	LD_50_: 4 (4–5)	Reduced activity, slowed breathing, convulsions, respiratory depression	[56]
Gelsenicine	Mice	Intraperitoneal injection	LD_50_: 0.165	Death	[57]
Gelsenicine	Mice	Subcutaneous injection	LD_50_: 0.1–0.2	Death	[28]
Koumine	Mice	Intraperitoneal injection	LD_50_: 99	Dyspnea and tonic convulsions before death	[58]
Gelsemine	Mice	Intraperitoneal injection	LD_50_: 56.2	Convulsions before death	[58]
N1-methoxygelsemine	Mice	Intraperitoneal injection	LD_50_: 63.1	Increased activity, convulsions before death	[58]
Kouminicine	Mice	Intraperitoneal injection	LD_50_: 2.83	Death	[58]
Kouminicine	Rats	Intravenous injection	LD_50_: 0.7	Tonic convulsion and death due to respiratory depression	[58]
Koumidine	Mice	Intraperitoneal injection	LD_50_ >125	Paralysis or death	[58]
Koumicine	Mice	Intraperitoneal injection	LD_50_ >125	Reduced activity, dyskinesia, and dyspnea	[58]
Gelsedine	Rats	Intraperitoneal injection	MLC: 0.1–0.12	Respiratory depression, convulsions	[59]
Gelsedine	Frogs	Injection to abdominal lymphatic sac	MLC: 20–30	Respiratory depression, convulsions	[59]
Gelsedine	Rabbits	Intravenous injection	MLC: 0.05–0.06	Respiratory depression, convulsions	[59]
Gelsedine	Dogs	Intravenous injection	MLC: 0.5–1.0	Respiratory depression, convulsions	[59]
Total alkaloids of GEB	Rabbits	Intravenous injection	MLC: 4–8	Decreased heart rate and respiratory depression before death	[60]
GEB alkaloids injection	Male rats	Intraperitoneal injection	LD_50_: 1.2 (0.8–1.7)	Decrease in quiet activities, dyspnea, convulsions before death, death within 3–12 h	[26]
GEB alkaloids injection	Female mice	Intramuscular injection	LD_50_: 1.5 (1.4–1.6)	Respiratory depression and convulsions before death	[26]
GEB alkaloids injection	Mice (male-to-female ratio: 1:1)	Intravenous injection	LD_50_: 1.56 (1.4–1.69)	Respiratory depression and convulsions before death	[26]
GEB alkaloids injection	Male rabbits	Intravenous injection	LD_50_: 76	Excitement, convulsions, and dyspnea before death	[26]

## Data Availability

Data are contained within the article.

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
