# Peer review of "Gelsemium elegans Benth: Chemical Components, Pharmacological Effects, and Toxicity Mechanisms"

_molecules, 2021, doi:10.3390/molecules26237145_

Round 1

Reviewer 1 Report

This manuscript reviews the alkaloid constituents of Gelsemium elegans Benth and their many bioactivities. The report is well written and easy to follow. The authors are thorough and concise in their summaries of the various pharmacological and toxic effects of the GEB alkaloids. GEB alkaloids are interesting potential therapeutics so this up-to-date review on their positive and negative activities is a welcome and helpful addition to the literature. Please address the following minor edits and suggestions:

Figure 1. change “comar” to “coma”

Line 70: please describe the structure activity relationship in more detail.

Table 3. can IC50 values or ranges of each study/compound also be included in the table?

Can a table be made to summarize the other activities with effective dose values (e.g. IC50 or MIC values)?

Author Response

Dear anonymous reviewer,

We are glad to receive the valuable comments and suggestions from you to our manuscript. Thank you very much for your kind consideration on this manuscript “Gelsemium elegans Benth: Chemical Components, Pharmacological Effects, and Toxicity Mechanisms”. Without your professional reviews, this manuscript would not be as smooth as what it is now. Thanks again for the revision opportunity very much!

We have amended the manuscript according to all the opinions, suggestions and comments. The responses to all the comments and suggestions are itemized as attached.

This is again to confirm that we have amended the manuscript according to all your suggestions. Thanks again for your quick processing and professional editing of this manuscript. What you have done will always be highly and greatly appreciated.

Any questions, we will be more than happy to answer.

Regards and Best wishes!

Youxiong Que, Wancai Que

2021-11-17

Reviewer 2 Report

The manuscript molecules-1432393 is a review that develop the subject of  compounds of the Gelsemium elegans Benth. It is an interesting and concise review. However I would like to address some comments.

The title of the manuscript claims that the manuscript overviews the chemical componets of GEB, or at least the vast majority of them. In the Abstract the readers discover that the study is focused on the indole
alkaloids. The last sentence of the Abstract claims "In-depth research on the chemical components ....". Also, in the main text of the manuscript, the authors are focused on the  indole alkaloids.

The authors should specify the limitations of their study not just L84 "However, further analyses of the specific mechanisms are still necessary." The readers expect to know more about the perspective of the studies that could advance the understanding of these specific mechanisms.

In my opinion some figures would be helpful to have a global image of mechanisms described in the main text.

L34 "Given its relatively high toxicity, the clinical application of GEB has been limited." The clinical application could be summarized more explicitly in the main text.

Also, L226 " Summary and Prospect" is a Conclusion section. This is a review, it is not necessary to high-light many times that L233 "Further research is needed to study...", L238 "which needs to be further studied". The manuscript should clearly presents what was done and to analyze the results already published - the strength and the limitations of the studied. If is suitable some statistical analyzes of the data would add value of the manuscript. A perspective, a personal point of view is welcome. 

The authors should provide the methods of selection of the references included in their analysis - the inclusion and exclusion criteria.

minor comments:

Please use a constant name " indole alkaloids" or " alkaloid components"

Some paragraph could be polish.

e.g. L33 "to dispel the wind, reduce swelling, remove poison, kill insects, and relieve itching."

Author Response

Dear anonymous reviewer,

We are glad to receive the valuable comments and suggestions from you to our manuscript. Thank you very much for your kind consideration on this manuscript “Gelsemium elegans Benth: Chemical Components, Pharmacological Effects, and Toxicity Mechanisms”. Without your professional reviews, this manuscript would not be as smooth as what it is now. Thanks again for the revision opportunity very much!

We have amended the manuscript according to all the opinions, suggestions and comments. The responses to all the comments and suggestions are itemized as attached.

Thanks again for your quick processing and professional editing of this manuscript. What you have done will always be highly and greatly appreciated. Any questions, we will be more than happy to answer.  

Regards and Best wishes!

Youxiong Que, Wancai Que

2021-11-17

Round 2

Reviewer 2 Report

Dear Youxiong Que, Wancai Que,

Thank you that finding my comments useful.

The manuscript is very much improved. However, I have to stresses that a review should be conducted by certain rules (e.g. PRISMA).

L38 "present study, after searching and reviewing in the databases of MEDLINE/PubMed, Google Scholar, CNKI, Scopus, and Web of Science, 60 papers tightly related with chemical components, pharmacological effects, and toxicity mechanisms were identified..."

It is an impressive search but please add the search terms and the date when the search was made. Please specify the number of the articles retrieved after the first search. Also, I assume that you and your colleagues are made a screening and selected the 60 references according to some specific reasons.

Author Response

Dear anonymous reviewer,

Again we are glad to receive the valuable comments and suggestions from you to our manuscript. Thank you very much for your kind words on our manuscript “Gelsemium elegans Benth: Chemical Components, Pharmacological Effects, and Toxicity Mechanisms”. I am so happy that with your professional reviews, this manuscript can be as smooth as what it is now.

The responses to all the comments and suggestions are itemized as attached.

This is also to confirm that I have double checked the language of the whole manuscript. Take this opportunity, thanks again for your professional review. I appreciate so much you have done for this manuscript.

Regards and Best wishes!

Youxiong Que, Wancai Que

2021-11-20
